# Personalized Nutritional Therapy Based on Blood Data Analysis for Malaise Patients

**DOI:** 10.3390/nu13103641

**Published:** 2021-10-18

**Authors:** Minoru Arakaki, Li Li, Toshiyuki Kaneko, Hiromi Arakaki, Hiromi Fukumura, Chihiro Osaki, Maki Yonamine, Yoshitaka Fukuzawa

**Affiliations:** 1Arakaki Plastic Surgery, 729 Uchidomari, Ginowan 901-2227, Japan; hiromi-a@cosmoe.ne.jp (H.A.); fukumura@arakakikeisei.com (H.F.); chihiro@arakakikeisei.com (C.O.); akcmac@gmail.com (M.Y.); 2QDU International Tele-Consultation Platform, Qingdao United Family Hospital, 319 Hong Kong East Road, Laoshan District, Qingdao 266102, China; lilidoc@126.com; 3Orthomolecular Nutrition Laboratory Inc., KYB Build. 3F, 1-6-13 Higashi, Sibuya-ku, Tokyo 150-0011, Japan; t.kaneko.19790402@gmail.com; 4Aichi Medical Preemptive and Integrative Medicine Center, Aichi Medical University Hospital, 1-1 Yazakokarimata, Nagakute 480-1103, Japan; yofuku@aichi-med-u.ac.jp

**Keywords:** nutrition, preventive medicine, personalized medicine, ortho-molecular nutrition, blood examination, anti-aging

## Abstract

As medical doctors, we routinely check patient blood chemistry and CBC data to diagnose disease. However, these data and methods of analysis are very rarely used to find pre-disease conditions or treat undiagnosed malaise. Masatoshi Kaneko Ph.D. found that many pre-disease conditions and types of malaise could be detected using his unique method of blood data analysis, and could also be treated using personalized nutritional therapy as an alternative to using drugs. The authors of this article introduce personalized nutritional therapy based on blood data analysis (Kaneko’s method), and present and discuss some clinical cases. In total, 253 pre-disease or undiagnosed patients were treated using this nutritional therapy approach, and most of them recovered from their chronic symptoms and pre-disease conditions. This novel nutritional therapy has the potential to help many presymptomatic and undiagnosed patients suffering from malaise.

## 1. Introduction

Advances in molecular nutrition have generated the concept of ortho-molecular nutritional therapy. Dr. Masatoshi Kaneko pioneered the concept of ortho-molecular nutritional therapy in Japan. Kaneko determined the optimal range (ideal standard values) for blood examination by analyzing more than 350,000 blood data sets. He found that the minimum deviation from the optimal ranges for blood data can be used to diagnose deficiencies in various nutrients, and that certain combinations of blood data indicate sub-optimal function of specific organs. For example, in the absence of liver or bone disease, a low alkaline phosphatase (ALP) level suggests zinc deficiency; a serum level of aspartate aminotransferase (AST) > alanine aminotransferase (ALT) indicates vitamin B6 deficiency; an increase in mean corpuscular volume (MCV) with low ferritin means weakness of cell membranes; and a reduction in blood urea nitrogen (BUN) without kidney disease means a low protein intake [1]. These physiological readings are never taught to the current generation of medical students or medical doctors. 

Kaneko also found that many symptoms indicated by blood data can be treated with nutrients according to his defined optimal ranges. He applied personalized nutritional therapy based on the blood data analyses of patients experiencing malaise and prescribed typical courses of personalized nutritional therapy. Using this method, various organ malfunctions in presymptomatic, undiagnosed patients can be effectively corrected with relatively fewer side effects compared to current conventional medicines [1,2,3]. Additionally, ortho-molecular nutritional therapy can reduce the side effects of strong medicines such as anti-cancer drugs by supporting normal organ function for cancer patients [4,5]. 

Using Dr. Kaneko’s pioneering techniques, the authors of this paper successfully treated 253 presymptomatic patients and patients with incurable illnesses by prescribing nutritional therapy. Patients were screened using the Anti-Aging QOL Common Questionnaire (AAQC-Questionnaire) promulgated by the Japanese Society of Anti-aging Medicine [6]. The results were impressive. The total severity score among patients ranged from 50 to 173 points and the mean value of total severity score was 90.56 points before nutritional therapy. After three months of nutritional therapy, the total severity score ranged from 45 to 153 points and the mean value decreased to 82.02 points. All the symptoms tended to decrease after the nutritional therapy, and the mean values for 23/26 of the physical-symptom-related questions and 15/16 of the mental-symptom-related questions showed statistically significant differences before and after three months of nutritional therapy.

Further investigation is needed to confirm the efficacy of Kanko’s method with a larger sample size and at a population level. Nevertheless, preliminary results show that this novel nutritional therapy has the potential to help many presymptomatic and undiagnosed patients suffering from malaise. 

The purpose of this article is to evaluate the efficacy of this unpublished but novel practical method of nutritional therapy, demonstrate the importance of the molecular function of nutrients, and to show how clinicians can incorporate nutritional therapy in their medical practices.

## 2. Materials and Methods

### 2.1. Patient Selection: A Total of 1021 Patients Who Visited our Outpatient Clinic from 2018 to 2020 Were Investigated Retrospectively 

A total of 253 patient candidates of the 1021 (including 15 end-stage cancer patients) who complained of malaise, had received more than 3 months of consistent nutritional therapy, and completed blood data analyses both before and after nutritional therapy were selected for this study.

Malaise is defined as more than 5 symptoms of which severity is 2-scale or more on the initial AAQC-Questionnaire (Figure 1).

Age range of participants: 19–78 years old. Sex distribution: 63 males and 190 females. The disproportionate number of female study participants (i.e., of the 1024 total participants, 738 were females vs. 286 males) reflects the disproportionate number of patients seeking care in a typical cosmetic surgery practice. 

### 2.2. Data Sampling

Patient histories were taken, and conventional blood-chemistry tests and complete blood counts (CBCs) were performed. The blood tests were conducted prior to commencement of nutritional therapy, repeated every 3 months during the intervention, and until the completion of the treatment. Diet questionnaire sheet and AAQC-Questionnaire were also taken before and after 3 months of nutritional therapy for food guidance and evaluation.

### 2.3. Nutritional Interpretation of Blood Data

According to the Kaneko’s method, the blood data were analyzed by nutritional interpretation. The nutritional target values and meanings are shown in Table 1. These target values were derived from a KYB Medical Service Inc. database (unpublished) data set of 35 million healthy Japanese blood test samples and were considered in the context of the theoretical values under physical conditions [1,2].

Target values were determined with consideration of the physiological state of the individual’s body. When patients present pathological states, parameters are masked, so we have to examine the variables’ data carefully in order to prevent misleading evaluations and conclusions. 

### 2.4. Personalized Nutritional Therapy (Kaneko’s Method)

According to the nutritional interpretation of blood data, necessary nutrients such as protein, vitamins, minerals, essential fatty acids, enzymes, and probiotics were prescribed for each patient. The dosage of each nutrient was adjusted according to the data from blood tests repeated every 3 months. 

This was in response to the emphasis placed by Kaneko on the importance of giving sufficient doses of specific nutrients to each patient.

Prior to starting nutritional therapy, the doctor explained the blood test results to the patient. After informed consent, the doctor wrote the prescription for the supplements and food change for the patient. Based on the doctor’s prescription, the nutritionist in charge gave specific and detailed guidance to the patient and checked up on the progress of the nutritional therapy monthly. After 3 months of nutritional therapy, the blood tests were repeated and the before and after biometrics were analyzed based on the guidance of the AAQC-Questionnaire. 

A brief explanation about the meaning of each target value and recommended supplementation and food guidance are represented as follows. 

#### 2.4.1. Total Protein (TP) and Albumin (Alb)

The target serum protein level is 7.5–8.0 mg/dL (Alb = 4.5 mg/dL).

When the TP and Alb levels are ≤7.0 mg/dL and ≤4.0 mg/dL, respectively, a low protein intake is suspected [2]. 

The main advice is to alter food intake to include more protein and amino acids.

Miso soup, tofu and natto with raw egg are the most recommended traditional Japanese foods for protein and amino acids. Soy protein powder is recommended as a supplement.

#### 2.4.2. BUN and Creatinine (Cre) 

The target BUN level is 20–22 mg/dL. In case of a low score for BUN (≤15 mg/dL) with a low aminotransferase level (AST and ALT, <15 U/L), low protein synthesis is suspected [1]. Vitamin B complex and zinc are recommended for elevating protein synthesis.

The target Cre level is 0.8–1.0 mg/dL. A low score for Cre (≤0.5 mg/dL) without kidney disease means a loss of skeletal-muscle mass. Protein powder or branched-chain amino acids (BCAAs) in combination with exercise are recommended.

#### 2.4.3. Total Cholesterol (T-Cho) and High-Density Lipoprotein Cholesterol (HDL)

The target value for T-Cho is 200–220 mg/dL, and the target value for HDL is 70–90 mg/dL. Cholesterol balance is very important, and the atherosclerosis index (AI = (T-Cho – HDL) ÷ HDL) should be 3.0 or lower [1]. A low cholesterol level (T-Cho ≤ 170 mg/dL) is a concern when combined with steroid hormone and vitamin D deficiency. Eicosapentaenoic acid (EPA) and docosahexaenoic acid (DHA) are recommended in order to elevate T-Cho and HDL. Vitamin D3 can be taken as a supplement. When T-Cho is high (≥250 mg/dL), a low-fat diet, exercise, and vitamin B3 are recommended before using statins. 

#### 2.4.4. Triglyceride (TG)

The target value for TG is 100–150 mg/dL. TG is a parameter of energy reserve that mainly reflects carbohydrate intake. Even if the TG level is higher than 150 mg/dL, we should not simply advise reducing carbohydrate intake. A change from high-glycemic index (GI) foods to low-GI foods is recommended. For example, whole grain rice and whole grain flour are preferable over processed or “white” options. A calorie-reduced diet is often too stressful for patients to maintain, so substituting certain foods has been found to be a better and more sustainable approach. Exercise is, of course, recommended for burning TG [1].

#### 2.4.5. AST, ALT and γ-Glutamyl Transpeptidase (γ-GTP)

The target values for AST, ALT and γ-GTP are 20–22 U/L. It is well known that AST > ALT indicates vitamin B6 deficiency and AST < ALT indicates fatty liver. Vitamin B complex supplementation is recommended in both cases. When γ-GTP is less than 15 U/L, low protein synthesis is considered [1,7]. Vitamin B complex with zinc supplementation is recommended. All these measurements should be taken in combination with those of the other parameters. 

#### 2.4.6. Alkaline Phosphatase (ALP)

The target value for zinc is 200–220 U/L (JSCC). Zinc is the active component of ALP, so when ALP is less than 150 U/L(JSCC), zinc deficiency is suspected [8]. Zinc supplement intake is recommended. Today, zinc can be directly measured through a blood examination, but ALP is still useful if zinc cannot be measured. 

#### 2.4.7. Lactate Dehydrogenase (LDH)

The target value for LDH is 200–220 U/L. A high LDH level indicates tissue or cell damage. When combined with other parameters, e.g., high LDH with a large MCV, this factor indicates weakness in red-blood-cell membranes caused by vitamin B12 and folic acid deficiency. A low level of LDH (≤150 U/L) indicates low energy production, which may be caused by vitamin B3 deficiency [1,9]. Vitamin B3, coenzyme Q10 and magnesium supplementation are recommended to elevate energy production.

#### 2.4.8. Ferritin

The target value for Ferritin is 80–150 ng/mL. Ferritin is a very sensitive indicator of anemia—more sensitive than Hb (hemoglobin). Although the target value for ferritin is 80 ng/mL or more, the mean value for most premenopausal adult Japanese woman is 20 ng/mL or less due to menstruation, and almost all adult Japanese women need heme-iron supplementation. On the other hand, the mean value for adult Japanese men is 150 ng/mL or more. Upon identifying a low ferritin level (≤100 ng/mL) in a middle-aged Japanese man, some bleeding or other form of blood loss is suspected, as is common with intestinal hemorrhages or repeated blood donation [1,10].

#### 2.4.9. MCV

The target value for the MCV is 90–92 fL. When the MCV is less than 85 fL, iron-deficiency anemia is suspected; on the other hand, when the MCV is more than 95 fL, cell-membrane weakness is suspected, as caused by vitamin B12 or folic acid insufficiency, and cholesterol deficiency is also suspected. Vitamin B12 and folic acid supplementation is recommended.

#### 2.4.10. Other Parameters

(a) Measuring blood sugar (BS) with insulin at the same time after meals is very useful for detecting non-diabetic hypoglycemia. When the BS level is less than 80 mg/dL and insulin is higher than 15 U/mL, non-diabetic hypoglycemia is suspected. Sugar and high-glycemic-index (GI) foods should be excluded from the diet, and low-GI foods are recommended, accompanied by the training of patients by a nutritionist in order to understand the GI value as well as the providing of a GI-food Chart to help patients with choice of low GI food. [1].

(b) Neutrophil–lymphocyte ratio (NLR). The NLR can be used as a biomarker for a cancer patient’s prognosis and nutrition assessment [11]. The NLR is also useful for determining autonomic nerve balance [12]. Neutrophil count (NEUT) = 70% and lymphocyte count (LYM) = 30% indicates the most stable state. NEUT > 80% with LYM < 20% indicates a sympathetic-dominant state, while NEUT < 60% with LYM > 40% suggests a parasympathetic-dominant state.

(c) Immunodeficiency. When immunodeficiency is suspected, such as in a cancer patient who has received chemotherapy and merged agranulocytosis, a hyperconcentrated vitamin C injection is useful. This can be used for recovery from exhaustion after hard physical work or strenuous physical activity, such as marathons, which cause oxidative stress. Glucose-6-phosphate dehydrogenase (G6PD) must be checked before therapy with the injection of hyperconcentrated vitamin C. Otherwise, hyperosmotic fluid injection may cause hemolysis and kidney damage in G6PD-deficient patients. For the prevention or reduction of the side effects of chemotherapy, a vitamin C 15–100 g/day DIV is necessary to maintain an effective plasma vitamin C concentration [13].

(d) Food guidance. Dietary counseling and food education are essential for maintaining good nutritional conditions for patients. Doctors usually advise a balanced diet to their patients, but it is difficult to concretely understand what a balanced diet is. To ascertain whether a patient’s diet is balanced or unbalanced, a 7-day diet questionnaire sheet is very useful. In the sheet, the nutrients are colored differently such as yellow for protein, red for carbohydrate and green for essential nutrients such as vitamins and minerals to visually demonstrate the balance of nutrients at the nutritional counselling. Wholegrain rice and buckwheat noodles are recommended as low-GI foods for Japanese people. Blue-backed fish, such as mackerel, saury and sardines, are good sources of ω-3 fatty acids (EPA/DHA). Tuna is not recommended because of its high mercury level. For best results, highly processed junk foods, especially those containing trans fats, must be eliminated from patients’ diets.

(e) Exercise. Exercise is very beneficial with regards to achieving superior results when used together with nutritional therapy. To improve the metabolism, we usually advised walking and yoga for the patients who complained of easy weight gain and cold intolerance. The patients who complained of insomnia, being a light sleeper and unrestful sleep were recommended an early bedtime and an early wake time and to soak up the morning sun in order to correct the circadian rhythm and get good sleep. Meditation with deep breathing was also advised for the patients who felt stress and complained of anxiety, fear and depression. 

### 2.5. Evaluation of Nutritional Therapy

The Anti-aging QOL Common Questionnaire, as issued by the Japanese Society of Anti-aging Medicine, was administered before and after the evaluation of nutritional therapy (Figure 1). Oguma et al. reported the efficacy of this questionnaire and several journals allowed its use as an evaluation tool [6,14,15,16]. Here, the severity of each symptom is evaluated on a 5-point scale (1: Never; 2: Not often; 3: Sometimes; 4: Often; 5: Always). The AAQC-Questionnaire sheet was given to patients at the counseling room before nutritional therapy as the initial information and it was then corrected and administered by doctors. At the end of the examination, a new AAQC-Questionnaire sheet was given to the patients to evaluate the results of the nutritional therapy. 

Statistical analysis was carried out for each symptom for comparison before and after nutritional therapy using paired *t*-tests (*n* = 255; * *p* < 0.05; ** *p*< 0.01). 

## 3. Results

### Anti-Aging QOL Common Questionnaire

The original Anti-aging QOL Common Questionnaire comprises 33 questions relating to physical symptoms and 21 questions relating to mental symptoms. Some questions were deemed incompatible with the goals of nutritional therapy, and some questions were found to be very similar and difficult for patients to answer. As such, a selection of 26 of the physical- and 16 of the mental-symptom-related questions were selected for the purpose of this study.

The total severity score ranged from 50 to 173 points and the mean value of total severity score was 90.56 points before nutritional therapy. After three months of nutritional therapy, the total severity score ranged from 45 to 153 points and the mean value decreased to 82.02 points, respectively.

The mean values of the degree of each symptom (*n* = 253) were calculated and compared before and after three months of nutritional therapy to ascertain the type of symptoms that can be treated by nutritional therapy. 

All the symptoms tended to decrease after nutritional therapy, and the mean values for 23/26 of the physical-symptom-related questions and 15/16 of the mental-symptom-related ones showed statistically significant differences from before to after three months of nutritional therapy (*t*-test; *n* = 253; * *p* < 0.05; ** *p* < 0.01) (Figure 2 and Figure 3).

Of the physical symptoms, an unhealthy skin condition, joint pain, and frequent urination tended to decrease but not in a statistically significant fashion.

Of the mental symptoms, feeling there was no purpose to live was not statistically significant.

## 4. Discussion

A nutritional therapy based on blood examination and analysis (Kaneko’s method) was evaluated in this study. The results show significant effects were achieved using the nutritional therapy based on personalized blood examinations and analyses, supplement prescriptions and dietary counseling.

In Japan, supplements are categorized as foods, so there has been thought to be no evidence to support supplementation as a medical treatment. However, advancements in bioscience have revealed the molecular mechanisms of action in nutrients, and as a result, the importance of nutrients is being readdressed by many researchers across the medical sciences. 

Ortho-molecular medicine, as established by Linus Pauling and Abram Hoffer, is a typical point of reference. 

In Japan, companies are obligated to ensure that all employees receive an annual health check, called the NINGEN DOCK. This compulsory, comprehensive annual health check for adults considered otherwise healthy has been part of the law since 1974 [17], and everyone aged 40 to 75 has had to receive a specified medical examination once a year since 2008 [18]. 

Kaneko focused on screening these blood data and made connections between the active mechanisms of specific nutrients through their enzyme activity. He hypothesized that enzyme activity has an optimal range, and a certain degree of deviation from this range indicates a presymptomatic condition. He theorized that the reference ranges were not useful for detecting individual nutritional deficiencies. 

The target values adopted in Kaneko’s method are not reference intervals currently approved by the IFCC or JSCC. However, Kaneko’s method, based on a large national blood data survey, has been able to decipher and theoretically estimate prophylactic thresholds for human health from cell functions or metabolism levels that fall under the thresholds for conventionally diagnosed physical conditions.

Kaneko’s method is mainly used to detect reduced enzyme activity or metabolism in patients and, more importantly, has been used to monitor nutritional therapy and correct dosages. After analyzing the blood data of more than 35 million individuals, Kaneko found that almost-unknown symptoms arise under presymptomatic conditions, and these can be effectively treated using nutritional therapy. 

To the surprise of the authors of this paper, as far as we are aware, very few studies have reported the detection of nutrient deficiencies as estimated from blood-chemistry data outside of Kaneko’s work.

One of Kaneko’s greatest contributions was to identify occult iron deficiency (iron deficiency without anemia). He reported many undiagnosed symptoms that arose before Hb changes that could be detected via the serum ferritin level and treated effectively by supplementation with heme iron [19].

The World Health Organization published the Vitamin and Mineral Requirements in Human Nutrition, 2nd edition, in 2005. In this report, they emphasized the importance of defining EARs (estimated average requirements), RNIs (recommended nutrient intakes) and Uls (upper limits of nutrient intake), which prevent vitamin and mineral deficiencies and avoid the consequences of excess. However, they also admit that these values are for populations, not for individuals, because the required nutritional dosages vary significantly according to age, gender, race, etc. [20].

Kaneko’s method is a personalized nutritional therapy approach based on the analysis of blood-test data. The target values and dosages of nutrients he set should be verified through further investigations. His settings are almost within the reference intervals between RNIs and Uls, and we have observed no serious side effects produced using this method to date. 

Recently, the optimal ranges for nutrients have been assessed by many study groups, academic societies and commercial-based institutes, and elements such as vitamins, minerals and fatty acids have been measured by blood tests [21,22,23,24]. 

However, it is still uncommon to find Kaneko’s method being used by practicing doctors. All medical doctors can read blood-test data to examine the pathological states of patients, but most doctors are not able to read blood data from a nutritional or physiological perspective. This is because they never learn the physiological meanings of blood chemistry data in detail during their medical school education in Japan. Medical students are only taught the pathological meanings of blood data and how to diagnose diseases.

In addition to these limitations in current medical education, medical students are taught to use medical drugs for correcting disease, and do not learn how to use nutrients to maintain health or regulate cell functions.

To provide a simple example, let us assume that there is a patient who is experiencing fatigue as a chief complaint, and their blood data show AST at 89 mg/dL, ALT at 78 mg/dL and γ-GTP at 230 mg/dL. 

When traditionally trained doctors see these blood data, they will easily diagnose alcoholic liver dysfunction.

However, if the blood data show AST at 15 mg/dL, ALT at 11 mg/dL and γ-GTP at 8 mg/dL, many may miss the low liver function caused by vitamin B6 deficiency [25]. 

As another example, when the ALP level is high in a child, doctors can easily identify it as a result of high bone metabolism, and when the ALP level is high in the elderly, they will suspect decalcification of bones. However, when the ALP level is low, they may miss the indication of zinc deficiency in the body [26].

The authors of this paper realize the importance of nutritional therapy for patients who are experiencing malaise under presymptomatic conditions. Patients with iron deficiency and a normal Hb level but a low ferritin level may feel general malaise due to the low mitochondrial function caused by low cytochrome activity [27]. They may, for example, not be able to open the lid of a jar because of muscle weakness caused by the low myoglobin activity and they may experience increased pigmentation caused by low catalase activity, because these molecules require heme iron as their active center. 

Since the condition differs from malaise, 15 cancer cases have been included in this report; all the patients had diagnosed stage four cancers, received chemotherapy and suffered from side effects.

The efficacy of high-dose vitamin C injections for advanced cancer patients is still considered controversial [28], but the authors’ clinical experiences show that it can reduce fatigue, pain and the side effects of chemotherapy. 

Personalized nutritional therapy based on blood data analysis (Kanko’s method) has been introduced and discussed. This method can be easily adopted by medical doctors, similarly to the current methods used for medical diagnosis and therapy for diseases. If many doctors adopt this unique method of nutritional therapy and utilize it in their daily medical practice, they could assist and bring relief to many presymptomatic patients suffering from undiagnosed symptoms. 

In this article, a small number of clinical cases are reported, so further investigations are needed to confirm the efficacy of this method. However, Kaneko’s method is unique and represents the first attempt at devising a personalized nutritional therapy based on blood analysis, so the authors will continue the investigations, and future studies will include a higher number of cases. This novel nutritional therapy has the potential to help many presymptomatic and undiagnosed patients suffering from malaise.

## Figures and Tables

**Figure 1 nutrients-13-03641-f001:**
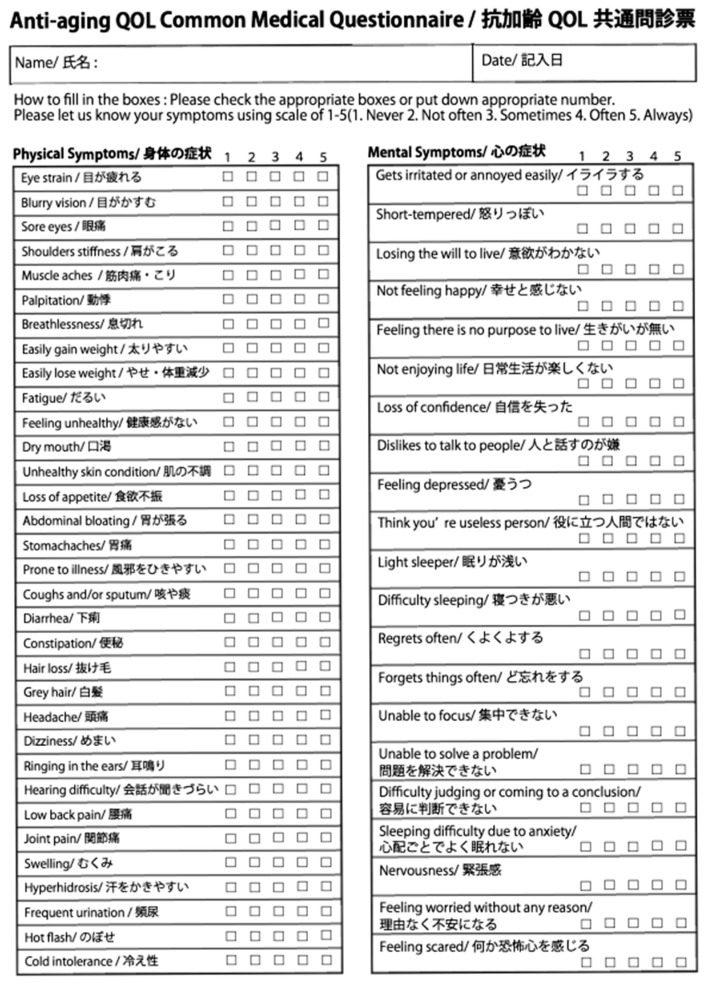
Anti-aging QOL Common Questionnaire. The questionnaire is composed of 33 questions relating to physical symptoms and 21 questions relating to mental and emotional symptoms.

**Figure 2 nutrients-13-03641-f002:**
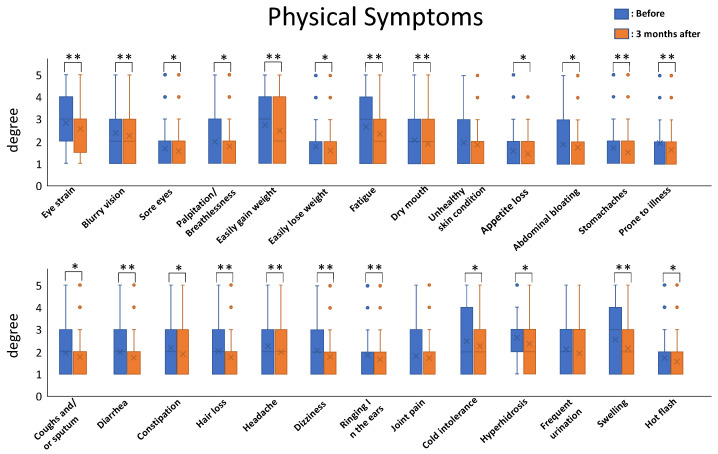
The mean values of degrees of physical symptoms. Blue bars indicate the mean scores before therapy, and orange bars indicate the mean scores after 3 months of nutritional therapy. Error bar indicates 2xSTDEV.S and Blue and Orange circles indicate outlier beyond 2xSTDEV.S. All symptoms tended to decrease, and the mean values for 23/26 of the physical symptoms showed statistically significant differences from before to after nutritional therapy (*t*-test: *n* = 253; *: *p* ≤ 0.05; **: *p* ≤ 0.01).

**Figure 3 nutrients-13-03641-f003:**
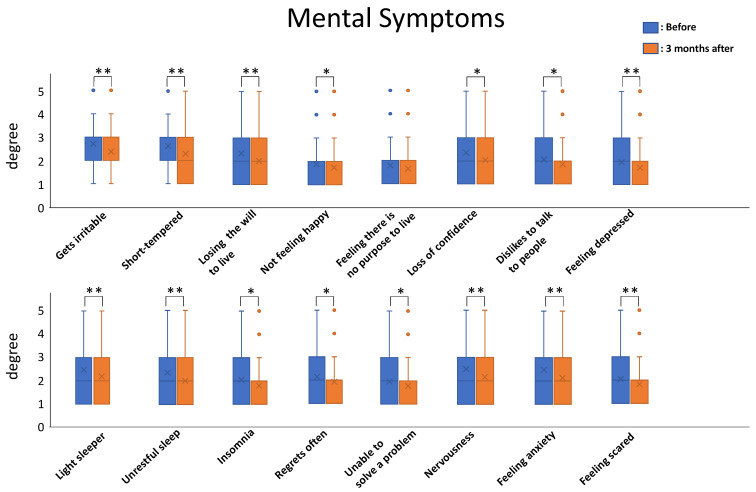
The mean values of degrees of mental symptoms. Blue bars indicate the mean scores before therapy, and orange bars indicate the mean scores after 3 months of nutritional therapy. Error bar indicates 2xSTDEV.S and Blue and Orange circles indicate outlier beyond 2xSTDEV.S. All symptoms tended to decrease, and 15/16 of the mean scores showed statistically significant differences from before to after nutritional therapy (*t*-test: *n* = 253 *: *p* ≤ 0.05; **: *p* ≤ 0.01).

**Table 1 nutrients-13-03641-t001:** Nutritional target values in blood test analysis.

Variable	Nutritional Target Value	Nutritional Interpretation
TP Alb BUN Cre T-Cho HDL TG AST ALT γ-GTP ALP LDH ChE CK Ferritin MCV	7.5–8.0 g/dL 4.5–5.0 g/dL 20–22 mg/dL 0.8–1.0 mg/dL 200–220 mg/dL 70–90 mg/dL 100–150 mg/dL 20–22 U/L 20–22 U/L 20–22 U/L 200–220 U/L 200–220 U/L 250–350 U/L 90–130 U/L 80–150 ng/mL 90–92 fL	Level of protein intake Protein sufficiency, level of protein synthesis in the liver Level of protein metabolism Parameter of skeletal muscle level Parameter of steroid-hormone metabolism Parameter of lipid metabolism Parameter of energy reserves AST > ALT = vitamin B6 deficiency AST < ALT = fatty liver state Parameter of protein synthesis Parameter of zinc sufficiency Energy metabolic level, low score = vitamin B3 deficiency Protein metabolism level, low score = vitamin B-complex deficiency ATP-synthesis level; low score = loss of skeletal muscle volume Parameter of anemia, more sensitive than Hb change 85 or less indicates iron deficiency 95 or more indicates cell membrane weakness

Abbreviations of variables: total protein (TP), albumin (Alb), blood urea nitrogen (BUN), creatinine (Cre), total cholesterol (T-Cho), high-density lipoprotein cholesterol (HDL), triglyceride (TG), aspartate aminotransferase (AST), alanine aminotransferase (ALT), γ-glutamyl transpeptidase (γ-GTP), alkaline phosphatase (ALP), lactate dehydrogenase (LDH), cholinesterase (ChE), creatine kinase (CK), mean corpuscular volume (MCV).

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
