# Peer review of "Personalized Nutritional Therapy Based on Blood Data Analysis for Malaise Patients"

_nutrients, 2021, doi:10.3390/nu13103641_

Round 1
Reviewer 1 Report
Personalized nutritional therapy based on Blood data analysis
Comments to authors
Overall, this is a very interesting article in the field of personalized nutrition, which highlights the importance of approaching nutritional guidance of patients away from the norm that one template of a diet plan can be efficient for all.
However, maybe the authors would like to consider the following points in order to address this issue properly:
Line 43-45: Must have a reference
Line 42 states “patients diagnosed with no abnormal conditions” while, Line 43 states “various organ malfunctions” Kindly specify if these are healthy patients or not.
Line 45: “Ortho-molecular nutrition can reduce the side effects of strong medicine[…] supporting optimal organ function” Maybe the authors should consider supporting this statement with appropriate references that will either explain the positive outcome of this approach or even better compare and provide evidence of its efficiency against a different nutritional approach. Also, what is considered optimal organ function? For a healthy population? For someone with a pathological condition? Please consider addressing the presentation of the sample population with more clarity.
Overall, since the major complaint of the patients is said to be malaise, maybe the authors should consider including that in the title as well.
Regarding the intervention:
- Were the patients advised to follow their regular lifestyle apart from the nutritional recommendations?
- Were any of the patients receiving any other (relevant perhaps) medications that could cause malaise or influence nutrient intake via drug-food interactions?
- Was the food intake of the patients monitored in order to evaluate adherence to the introduced nutritional approach?
- Line:84-86. Kindly consider clarifying if the patients were introduced to food options or food supplements.
Paragraph 2.4: Perhaps the authors could consider including the regular standard approach taken and then address the alternative nutritional approach introduced. This could either come from empirical data which is essential to be documented as well or from published data available for the authors’ region.
Regarding the tool of evaluation:
The Anti-aging QOL Common Questionnaire:
- Is this a validated tool for the population included in this study? (since there are patients <18 this might be a problem for the communication of the questions as well as the interpretation of results)
- Why was this tool selected for the evaluation of outcomes?
- A few extra lines on the description of the evaluation process might also be helpful, for instance, was the questionnaire given to the patients to answer on their own time, or was it administered by the doctors at the end of the examination?
Regarding the presentation of results:
It is not very clear if all the symptoms were present in all the patients before the intervention and this would be very helpful information for the reader. Kindly consider presenting this either here or in the description of the population before the intervention as mentioned in previous comment.
Regarding the discussion:
Kindly consider including the appropriate references throughout the discussion section.
I feel that the empirical point of view of the authors is essential to this work and should be highlighted in order to push further for the investigation of personalized nutrition approaches as well as the education level of medical doctors in the setting of nutrition and even look deeper in the cooperation (or lack thereof) with a nutritionist. However, it is essential to provide evidence of this approach being efficient in the comparison with other approaches previously employed in either research or common practice. On that note, although I consider this work to be very interesting and even necessary to be present in the available literature, I urge the authors to be clear and precise in their suggestions.
Author Response
Response to Reviewer 1 Comments
However, maybe the authors would like to consider the following points in order to address this issue properly:
1) Line 43-45: Must have a reference
Response: Author added 2-references
2)Line 42 states “patients diagnosed with no abnormal conditions” while, Line 43 states “various organ malfunctions” Kindly specify if these are healthy patients or not.
Response: Author change the sentence to “various organ malfunctions under presymptomatic condition”
3)Line 45: “Ortho-molecular nutrition can reduce the side effects of strong medicine[…] supporting optimal organ function” Maybe the authors should consider supporting this statement with appropriate references that will either explain the positive outcome of this approach or even better compare and provide evidence of its efficiency against a different nutritional approach. Also, what is considered optimal organ function? For a healthy population? For someone with a pathological condition? Please consider addressing the presentation of the sample population with more clarity.
Response: Author added 3 references and Change words “optimal organ function” to “normal organ function”
4)Overall, since the major complaint of the patients is said to be malaise, maybe the authors should consider including that in the title as well.
Response: Author Change the title to “Personalized nutritional therapy based on Blood data analysis for Malaise patients.”
5)Regarding the intervention:
Were the patients advised to follow their regular lifestyle apart from the nutritional recommendations?
Response: Yes. We recommend exercise, Yoga, aroma therapy etc.
Were any of the patients receiving any other (relevant perhaps) medications that could cause malaise or influence nutrient intake via drug-food interactions?
Response: Yes. Some patients took Pychotropic drags, Anticoagulant or Anticancer drags. Of course we consulted their attending doctors and got permission to use supplements such cases.
Was the food intake of the patients monitored in order to evaluate adherence to the introduced nutritional approach?
Response: Yes we do.
Line:84-86. Kindly consider clarifying if the patients were introduced to food options or food supplements.
Response: Some explanation is added in the article about food guidance.
Paragraph 2.4: Perhaps the authors could consider including the regular standard approach taken and then address the alternative nutritional approach introduced. This could either come from empirical data which is essential to be documented as well or from published data available for the authors’ region.
Regarding the tool of evaluation:
The Anti-aging QOL Common Questionnaire:
- Is this a validated tool for the population included in this study? (since there are patients <18 this might be a problem for the communication of the questions as well as the interpretation of results)
Response: 2 cases are excluded from this study under age of 18.
- Why was this tool selected for the evaluation of outcomes?
Response: Some explanation is added in the article about evaluation tool.
- A few extra lines on the description of the evaluation process might also be helpful, for instance, was the questionnaire given to the patients to answer on their own time, or was it administered by the doctors at the end of the examination?
Response: Some explanation is added in the article about administration of Anti-Aging QOL Common Questionnaire.
Regarding the presentation of results:
It is not very clear if all the symptoms were present in all the patients before the intervention and this would be very helpful information for the reader. Kindly consider presenting this either here or in the description of the population before the intervention as mentioned in previous comment.
Response: Some explanation is added in the article about determination process of Malaise patients.
Regarding the discussion:
Kindly consider including the appropriate references throughout the discussion section.
Response: Some references are added to discussion part.
I feel that the empirical point of view of the authors is essential to this work and should be highlighted in order to push further for the investigation of personalized nutrition approaches as well as the education level of medical doctors in the setting of nutrition and even look deeper in the cooperation (or lack thereof) with a nutritionist. However, it is essential to provide evidence of this approach being efficient in the comparison with other approaches previously employed in either research or common practice. On that note, although I consider this work to be very interesting and even necessary to be present in the available literature, I urge the authors to be clear and precise in their suggestions.
Response: Thank you for your review and advice very kindly.

Reviewer 2 Report
- The Introduction section is too short and should be added more information and the aim of the study should also be provided.
- Please explain why the sampled sex ratio vary so much.
- The number of clinical cases are too small.
- The description of conclusion is not clear enough.
Author Response
1.The Introduction section is too short and should be added more information and the aim of the study should also be provided.
Author added aim of the study and results of investigation and some references.
2.Please explain why the sampled sex ratio vary so much.
Because Author's Anti-Aging Clinic is based on Cosmetic Surgery originally so 80% of patients are Female.
3.The number of clinical cases are too small.
Authors know it, this is the previous report of this unique nutritional therapy and we are preparing next report with further investigation and higher number of clinical cases.
4. The description of conclusion is not clear enough.
More detail description and conclusion is added to discussion part.

Round 2
Reviewer 1 Report
The authors have addressed all of the initial recommendations.
I would suggest the following minor considerations:
- Maybe the authors would like to include a short clarification regarding the recommendation of exercise to the patients. Since the group analyzed in this work is described as presymptomatic patients with the primary complaint being malaise it would be helpful for the reader to understand what kind of exercise was suggested in order to further evaluate the outcomes of this work.
- Lines 193-194: The authors may consider including if the recommendation for low GI food was accompanied by a "training" of the patients in order to understand the GI value or if the patients were introduced to specific food choices.
- In relation to the previous comment, when applicable please consider including a short description of the intervention process under each target (were the patients given say a list of food choices or training that allowed them to make well-advised choices).
- Please revisit the format of the references at the end as they seem to have some differences.
This is a great introduction to the field of how personalized nutrition can aid in the prevention and management of malaise and the utilization of several tools available to health providers for the cause is well highlighted.
Reviewer 2 Report
There are still many errors in the English grammar and sentence pattern of the entire article that need to be improved. The answer to the reviewer’s questions is too brief and the English narrative need to be improved.
Author Response
Please see the attachment

This manuscript is a resubmission of an earlier submission. The following is a list of the peer review reports and author responses from that submission.
Round 1
Reviewer 1 Report
The issue of personalization of drug therapy and nutritional support is extremely important and the scientific community and practitioners should be expected to increasingly implement this approach in health care. However, the manuscript sent for review does not live up to the promises made in the title and abstract and does not meet the basic requirements for scientific papers. The introduction provides a weak scientific foundation of the the approach, number of study participants inconsistent with the abstract, the patient group not properly described, rationale for treatment criteria embarrassing, experiment vaguely documented, low-level discussion with residual references, including those not relevant to the discussion or not credible. I suggest rethinking the research concept, strategy and inference from the results. In this form the paper is not suitable for publication in Nutrients.
Reviewer 2 Report
To the authors
Greetings!
I thoroughly read your manuscript with interest. Your paper introduces a particular health care approach of diagnosing a number of conditions at their preclinical phase and treating them using specific nutritional therapies.
I have a number of preoccupations that are highlighted in the following lines.
- Paper editing is needed: it should undergo proofreading and appropriate editing work by health science editing service provider. Scientific English writing has syntax rules that should be used and, more importantly, medical technical terms must be used instead of general terms. All sections of the manuscript, from the abstract to the discussion, have numerous English syntax problems that need to be addressed. They should be fully revised.
- After reading this manuscript a number of times, my impression is that the objective remains unclear, as the report includes various data on different conditions. Normally, authors should focused on one or two conditions for a manuscript.
- Throughout the manuscript, authors used abbreviations of names of several biomarkers in blood, which is uncommon in scientific writing and publishing. For example, T-Cho, PT, AST, ALT, MCV ... and inappropriate terms such as "Crea" are used here and there in the paper. An abbreviation is always used after the full term or name is given. A scientific paper should be readable and understandable by any person; thus using only abbreviations makes the sentences difficult to understand for those who are unfamiliar with the field.
- The use of the term 'degree' for symptoms is not appropriate; 'severity' should be used instead.
- On page 4, last paragraph (other parameters), you mention blood sugar and there is statement saying that sugar should be excluded in case of hypoglycemia. Sugar does not cause hypoglycemia; thus, your statement is misleading and creates confusion.
- In the results section, you present in Figures data where statistical significance are shown; however, in the Methods section, there is no mention of statistical analysis and tests used for comparisons .
- In the Methods section, you wrote there were 255 patients whose clinical and laboratory data were used in the study; I wonder why you picked up separately two patients as clinical cases in the Results section. If data from all 255 patients were analyzed, is there a reason for selecting those 2 cases?

Reviewer 3 Report
Imroving nutrition to prevent and treat diseases has been considered valuable for decades. Authors expand this notion by recommending nutrient supplementation based on blood analysis. Conceptually this is a major step, but its practical use remains unknown. Authors studied 2 patients, one of whom died within year and other one showed improvement.. Furthermore, it is difficult to determine the dose of the supplement needed because in the main study the age range varied from 15--78 years. Table 1, some of the target ranges (eg 20-22) are very tight and not realistic to attain. Despite these and some other shortcomings of the paper, I firmly believe that nutrient supplement based on blood analysis will help in the treatment of some diseases and support further reasearch in this area. I encourage the authors to pursue it further.